# A Novel Color-Based Segmentation Method for the Objective Measurement of Human Masticatory Performance

**Luca Aquilanti** [1] , **Lorenzo Scalise** [2] , **Marco Mascitti** [1] , **Andrea Santarelli** [1,3,*] ,
**Rachele Napolitano** [2] , **Lorenzo Verdenelli** [2] and **Giorgio Rappelli** [1,3]

1 Department of Clinical Specialistic and Dental Sciences, Polytechnic University of Marche, Via Tronto 10, 60126 Ancona, Italy; l.aquilanti@pm.univpm.it (L.A.); m.mascitti@pm.univpm.it (M.M.); g.rappelli@staff.univpm.it (G.R.)

2 Department of Industrial Engineering and Mathematical Sciences, Polytechnic University of Marche, Via Brecce Bianche, 60131 Ancona, Italy; l.scalise@staff.univpm.it (L.S.); r.napolitano@staff.univpm.it (R.N.); l.verdenelli@pm.univpm.it (L.V.)

3 Dentistry Clinic, National Institute of Health and Science of Ageing, IRCCS INRCA, Via Tronto 10, 60126 Ancona, Italy

* Correspondence: andrea.santarelli@staff.univpm.it; Tel.: +39-071-220-6226

**Abstract:** The aims of this study were to propose an automatic color-based segmentation method to separate mixed and unmixed colors of images that were derived from the application of the two-color chewing-gum mixing test and to determine the validity of this method in the assessment of masticatory performance (MP). Fifty young adults (mean age: 24.3 ± 2.7 years) were enrolled in the study. Each participant chewed a double-colored chewing gum for 5, 10, 20, 30, and 50 masticatory cycles. Boluses were collected and flattened. Both sides of each bolus were photographed, and images were processed using a novel k-means clustering method. The specimens corresponding to 20 masticatory cycles were re-analyzed by the same investigator in order to evaluate the intra-rater reliability and by a second investigator to assess the inter-rater reliability. To assess the test–retest reliability, 25% of the participants performed a second test with 20 chewing cycles. Each bolus was subjectively scored as either poorly, moderately, or highly mixed by an investigator to assess the construct validity. The percentage of mixed colors in the samples increased with an increase in the number of strokes. Significative differences were detected when varying from 5 to 10 strokes, from 10 to 20 strokes, and from 30 to 50 strokes ($p < 0.05$). The Pearson correlation coefficient explained these relations (r = 0.78, $p < 0.05$). The interclass correlation coefficient (ICC) showed a good correlation concerning both the intra- and inter-rater reliability (r = 0.85 and r = 0.77, respectively) and an excellent test–retest correlation (r = 0.93). The subjective assessment was coherent with the digital one. The proposed digital method was proved to be able to automatically quantify the percentage of the mixed color area by providing quantitative data with minimal human interaction.

**Keywords:** mastication; software; chewing gum; masticatory performance

## 1. Introduction

Human mastication is a complex biomechanical process that is aimed at properly preparing food for swallowing and digestion [1]. Quality of life, general health, and social relations could be influenced by impaired mastication. Furthermore, the assessment of mastication could also be able to provide information about the efficiency and status of many oro-facial structures [2].

During their daily office practice, clinicians may be required to evaluate patient mastication. Thus, it is necessary to have an objective method that is able to assess it. The latter not only should aim at evaluating oral function but also at providing information about patient disability and a real indication for prosthetic rehabilitation. Over the past few years, several methods have been proposed in order to assess masticatory performance (MP). MP is defined as the individual ability to grind a test food after a fixed number of chewing strokes [3]. Most of the studies on MP used comminution tests in order to study the particles of both artificial and natural food by sieving the comminuted food [4]. Even though these systems are considered to be the gold standard in assessing MP, patient discomfort and high costs hinder their use. Besides the need for expensive and special equipment, particles of the specimens have to be completely removed from the mouth after the comminution procedure, which could be very difficult in the case of small fragments. Furthermore, dysphagic patients risk aspirating these particles [5]. Previous studies showed that the results of color-mixing ability tests highly correlate with those of comminution ones in assessing MP and proposed that color-mixing tests should be used on patients with chewing deficiencies [6,7].

In 1991 and 1995, Liedberg and Öwall proposed and described for the first time a new method for the study of MP. This test consisted of the chewing of two-colored chewing gum for 10, 20, 40, 60, 80, and 100 strokes. The specimens were then scored visually into a 1 to 5 color mixture scale, in accordance with a reference scale established in pilot tests [8,9]. In 1999, Prinz understood the need for simple objective tests for the assessment of oral function. Chewing gum containing two different colors was chewed. Once the bolus was removed from the oral cavity, it was flattened, and a digital image was taken. It was concluded that flattening the bolus before the assessment provided a more reliable evaluation than observing the bolus in its raw state. Nevertheless, the subjective evaluation had similar accuracy to image processing and the author concluded that subjective assessment was enough [10]. In 2007, Schimmel et al. tested the reliability and reproducibility of quantitative data obtained from the two-color chewing gum test and found that the visual assessment of the specimens was less reliable than computerized analysis. The authors concluded that the digital evaluation of the color mixing degree of two-color chewing gum was a precise and reliable method to assess MP [11]. As the computerized image analysis was carried out using a commercially available software package, in 2013, Halazonetis et al. proposed novel software for the quantitative assessment of MP [12]. The two-color chewing gum test and the dedicated software were validated in 2015 [13].

Even though the method proposed by Schimmel et al. is considered to be the gold standard among mixing ability tests, the digital analysis of boluses derived from its application is still controversial. Different software packages were proposed in order to analyze the specimens, highlighting the great variability of those methods and the need to find the best possible one to use both in clinical and research settings [12,14–18]. Unfortunately, the software may introduce errors due to the manual resizing and segmentation of bolus images and the difficulty in distinguishing hue variations between the gum and the background, especially for well-mixed chewing gum. Extracted data highly depend on the accuracy of the segmentation process, which is an essential step in computer vision and in automatic pattern recognition based on image analysis [19]. Clustering systems could provide high discriminative power that can discern different color regions in each image using color information. This could allow for quantifying the degree of color mixing depending on the number of chewing cycles or different dental patterns. The aims of the present study were to propose an automatic color-based segmentation method to separate mixed and unmixed colors of images derived from the application of the two-color chewing gum mixing test and to determine the validity of this method for the assessment of MP.

## 2. Materials and Methods

The study was performed on a sample of 50 subjects that were enrolled as students at the Dental School of Marche Polytechnic University, Ancona, Italy, from March 2019 to May 2019. They were young adults (mean age: 24.3 ± 2.7 years), homogenous for sex (25 males and 25 females), fully dentate

(28 teeth totally, third molars were not considered, 14 occluding pairs), in good general health condition, with a decay missed filled teeth (DMFT) less than 4, without temporomandibular joint (TMJ) disorders and in angle class I. They were taken as the reference population and were recruited for their similar oral conditions. All participants volunteered for this study. No sample size calculation was made, but a sample size that was 2.5-fold greater than those used in the studies of Schimmel et al. and Yousof et al. was enrolled [11,20]. The study was performed in full accordance with the World Medical Association Declaration of Helsinki and it was approved by the Local Review Board of the Dental School (ODO-ID062019). All the tests were performed with the informed and written consent of each subject and in accordance with the abovementioned principles.

The masticatory test was performed using the two-colored chewing gum mixing test as described by Schimmel et al., which involves the use of two-colored chewing gum (Hue-check Gum®, Orophys GmbH, Muri b. Bern, Switzerland) [8,10,11,13]. Each sample was chewed for 5, 10, 20, 30, and 50 chewing cycles. A rest interval of 1 min was set between each masticatory test in order to avoid muscle strain. Briefly, blue and pink gums were roughly stuck together using moderate strength. Specimens were placed on the tongue of the participants who were asked to chew in their usual way. Participants were allowed to change the chewing side during the test. An operator counted the masticatory strokes and at the defined chewing cycle, the operator asked the participants to stop. Boluses were retrieved from the mouth of the participants and gently dabbed with a paper towel in order to eliminate the possible excess of saliva. Specimens were collected by inserting them between two sheets of transparent plastic and flattened to a standard thickness of 1 mm. Figure 1 shows the used chewing gum and the boluses after 5, 10, 20, 30, and 50 masticatory cycles.

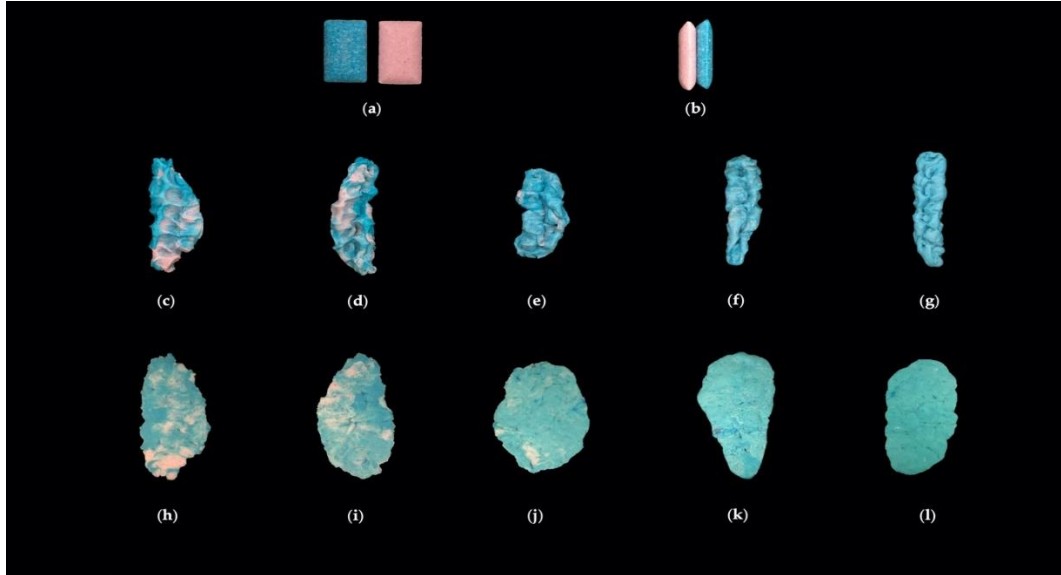

**Figure 1.** Chewing gum specimens: (**a**) the two colored chewing gums; (**b**) the two colored chewing gums stuck together; (**c**–**g**) chewing gum boluses after 5, 10, 20, 30, and 50 masticatory cycles, respectively; (**h**–**l**) flattened chewing gum boluses after 5, 10, 20, 30, and 50 masticatory cycles, respectively.

Both sides of each bolus were photographed, and, one side at a time, all images were post-processed using the segmentation algorithm developed by the Department of Industrial Engineering and Mathematical Sciences of the Polytechnic University of Marche, Ancona, Italy. The algorithm automatically analyzes the images and gives as output as the percentage of the mixed portion of the sample. The proposed algorithm exploits the k-means clustering method and the analysis is carried out by measuring the areas of different colors in pixels.

In the laboratory tests, accurate digitization of each sample was performed. The procedure was divided into two main phases:

1. An acquisition phase, in which the images of both sides of the boluses were captured using a micro-camera (FREDI HD MINI WIFI 1080P, Shenzhen Jinbaixun Technology Co., Ltd., Shenzhen, China) put at a standard distance of 10 cm. Each wafer was acquired on a dark background in order to make it easier to separate the different colors.

2. A processing phase, which was necessary to classify an image into different color groups. A PC (Intel® Core TM i7-6700HQ, 2.6 GHz, 16.0 GB) with Windows 10 Home was used as the electronic equipment to process the acquired data and MATLAB® R2019b (MatLab, MathWorks, Natick, MA, USA) software to enable the color-based segmentation.

K-means clustering was the method adopted for image processing in order to segment the mixed and unmixed areas of the chewing-gum. This method classifies each pixel in a region, according to its properties, such as color, intensity, or texture, to distinguish it from adjacent regions. For each region, called a cluster, it is possible to define k centroids placed as far as possible away from each other [21].

Each point belonging to a given data set is associated with the nearest centroid until no point is pending. When all objects are assigned to the initial group centroids, this step is executed to recalculate the positions of k new centroids as the barycenter of the clusters. After these k new centroids are found, a new binding needs to be done between the same data set points and the nearest new centroid. Because of this loop, it is possible to notice that the k centroids change their location systematically until the centroids do not move anymore. The goal of the algorithm is to minimize the total intra-cluster variance; this is equivalent to minimizing the pairwise squared deviations of points in the same cluster. A scheme of the implemented algorithm is shown in Figure 2.

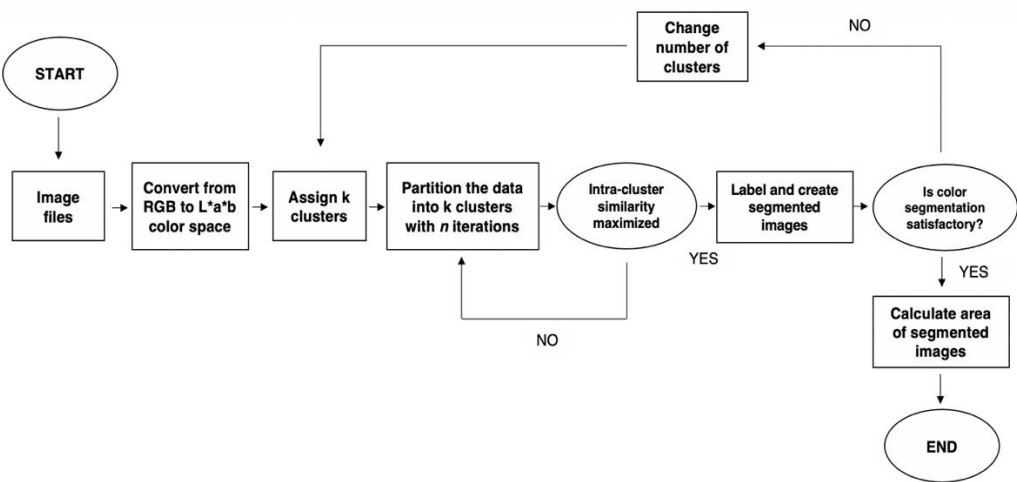

**Figure 2.** Block diagram of the implemented algorithm.

The areas of each chewed gum were divided into a set of four classes according to the pink, blue, mixed-color, and background areas. At the end of the analysis, the proposed software provided the percentage of the chewing gum mixed portion, discriminating the different MPs of the subjects (Figure 3). In particular, a higher percentage indicated a higher MP.

The specimens corresponding to 20 masticatory cycles were re-analyzed by the same investigator 1 week after the first set of measurements in order to evaluate the intra-rater reliability. All the images were also analyzed by a second investigator to assess the inter-rater reliability. Twenty-five percent of the participants ($n = 12$) were asked to repeat the test at 20 chewing cycles 1 week after the first test in order to assess the test–retest reliability. In addition, each specimen was classified visually and subjectively into poorly, moderately, or highly mixed categories by an investigator in order to assess the construct validity, as suggested by Montero et al. [17].

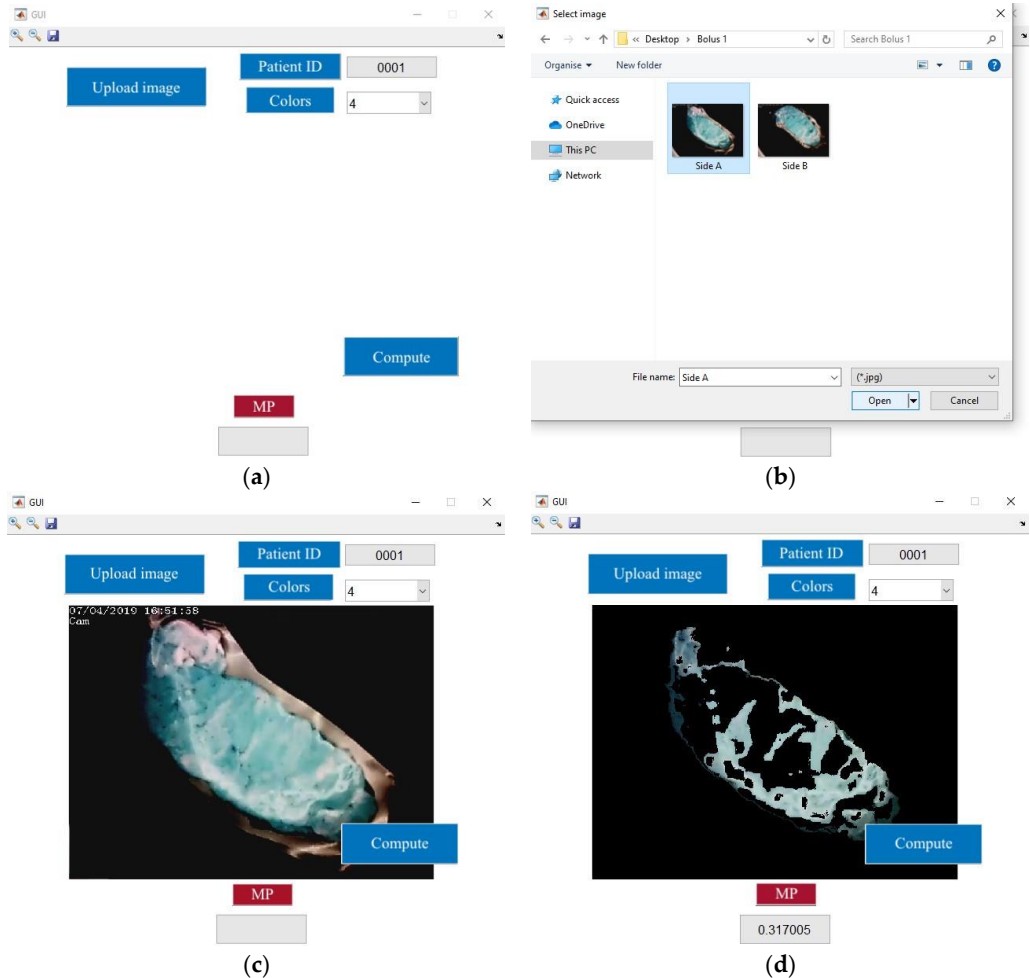

**Figure 3.** Digital analysis of the samples: (**a**) first view of the scene; (**b**) the user has to upload the image for analysis (one side of the bolus per time); (**c**) once the image is uploaded, it is ready to be analyzed; (**d**) the software automatically analyzes the image and gives as output the percentage of the mixed portion of the sample.

The statistical software R (R version 3.5.3 (11 May 2019)—"Great Truth," Copyright$^{©\ 2020}$ The R Foundation for Statistical Computing, Vienna, Austria) was used to perform the statistical analysis. The MP values were analyzed using the Pearson correlation coefficient in relation to the number of masticatory strokes. Group comparisons were performed using two-sample *t*-tests with unequal variances and Mann–Whitney tests. The similarity between the test samples of intra-rater, inter-observer, and test–retest reliabilities were evaluated using the interclass correlation coefficient (ICC) in a two-way mixed-effect model, with absolute agreement. According to Koo and Li, values less than 0.5, between 0.5 and 0.75, between 0.75 and 0.9, and greater than 0.9 are indicative of poor, moderate, good, and excellent reliability, respectively [22]. A value of $p < 0.05$ was considered statistically significant.

## 3. Results

The study was performed on a sample of 50 subjects who were enrolled at the Dental School of Marche Polytechnic University, Ancona, Italy, from March 2019 to May 2019. They were young adults (mean age: 24.3 ± 2.7 years), homogenous for sex (25 males and 25 females), fully dentate (28 teeth totally, third molars were not considered, 14 occluding pairs), in good general health condition, with a DMFT of less than 4, without TMJ disorders, and in angle class I. Table 1 summarizes the sociodemographic and clinical data of the study sample.

**Table 1.** Sociodemographic and clinical variables of sample size ($n = 50$). Data values are expressed as the number of participants for sex and mean and standard deviation (SD) for all other values.

| Sociodemographic Data | Value |
|---|---|
| Sex | |
| Male | 25 |
| Female | 25 |
| Age | 24.3 ± 2.7 |
| Clinical Data | |
| Natural teeth | 28 ± 0.0 |
| Occlusal units | 14 ± 0.0 |
| Filled teeth | 1.2 ± 1.0 |

A total of 250 specimens were obtained from the subjects enrolled in the present study. Both sides of the images of the samples, 500 images in total, were analyzed. It was possible to estimate the mixed areas of the chewing gums. Each color area was returned as the sum of the partitioned pixels and, for each participant, the mixed fraction of the two colors after 5, 10, 20, 30, and 50 chewing cycles was calculated.

The acquired data and their elaboration through the K-means method were used in order to assess the differences between the MPs of the 50 participants at 5, 10, 20, 30, and 50 chewing cycles. The results are summarized in Table 2.

**Table 2.** Masticatory performance (MP) values according to the number of chewing cycles. The results are expressed as the mean and standard deviation (SD).

| Chewing Cycles | 5 | 10 | 20 | 30 | 50 |
|---|---|---|---|---|---|
| MP | 0.228 ± 0.122 | 0.319 ± 0.139 | 0.443 ± 0.138 | 0.466 ± 0.111 | 0.830 ± 0.247 |
| Sex | | | | | |
| Male | 0.259 ± 0.110 | 0.352 ± 0.129 | 0.470 ± 0.157 | 0.466 ± 0.147 | 0.880 ± 0.237 |
| Female | 0.197 ± 0.128 | 0.286 ± 0.145 | 0.416 ± 0.112 | 0.466 ± 0.059 | 0.780 ± 0.252 |

No significant difference was observed between MPs of the male and female populations at 5, 10, 20, 30, and 50 chewing cycles ($p > 0.05$). A higher number of masticatory cycles corresponded to a higher value of MP. Considering the 50 participants and the different numbers of chewing cycles, the percentage of mixed colors in the samples increased with the increase in the number of strokes (Figure 3). The MP increase was statistically significant when varying from 5 to 10 cycles, from 10 to 20 cycles, and from 30 to 50 ($p < 0.05$). The MP increase was not statistically significant ($p > 0.05$) between 20 to 30 masticatory strokes. A ceiling effect was detected at 50 masticatory cycles in 22 participants, thus not allowing for distinguishing between different MPs. The Pearson correlation coefficient explained the relationship between the number of chewing cycles and MP (r = 0.78, $p < 0.05$).

The ICC results showed good correlations for both the intra- and inter-rater reliabilities (r = 0.85 and r = 0.77, respectively). The ICC showed an excellent test–retest correlation of the proposed software (r = 0.93). The proposed method was found to have adequate construct validity as statistically significant differences were detected between the MPs of the three different groups classified subjectively into poorly, moderately, and highly mixed ($p < 0.05$).

## 4. Discussion

In the present study, a novel method for the assessment of MP was proposed and described. MP was evaluated using a sample constituted by fully dentate young adults who chewed a two-colored chewing gum for different numbers of chewing cycles. Previous studies stated that a reduction in MP is more pronounced in patients with less than 20 teeth. In this report, 50 fully dentate participants were enrolled to determine the validity of the proposed method as they were considered ideal chewers [23,24]. They were fully dentate, in angle class I, with a DMFT < 4, and with no TMJ disorders. The choice

of a sample with these characteristics was considered crucial for the validation of the proposed software [10,12,16,18,25,26].

A recent systematic review stated that no established method for the clinical assessment of MP with a high level of evidence is available. Furthermore, all the proposed assessment methods reported in the literature need lab-intensive equipment, such as digital image software or sieves [25]. In 2018, Buser et al. tested the accuracy of a custom-built smartphone application, trying to overcome the need for specific and expensive instruments. Unfortunately, further development of that proposed application is not likely to happen [27]. In the present study, MP was evaluated using the k-means clustering technique. For each chewing gum wafer, mixed areas were evaluated. This clustering system using color information provided a high discriminative power for regions present in each image, reducing the errors due to manual segmentation. In particular, this method expressed MP as the percentage of mixed areas, where an MP of 1 indicates an optimal MP and an MP of 0 indicates the total absence of it.

Using the k-means clustering method, the MP was evaluated for each bolus by estimating the mixed and unmixed areas. This clustering system is able to provide high discriminative power for the regions present in each image using color information. Moreover, a reduction of the errors due to manual segmentation could be achieved. A significant increase in color mixing was observed in accordance with the different numbers of chewing cycles, as demonstrated in previous studies [13,28]. MPs corresponding to 5, 10, 20, 30, and 50 chewing strokes were compared. According to previous studies [10,28–31], a significant increase in color mixing depends on the number of chewing cycles, where 20 masticatory cycles were enough to determine the different MPs of subjects with different dental statuses. The variability between the MPs for different numbers of masticatory cycles, as shown in Figure 4, was in accordance with this datum, showing that 20 chewing strokes could be considered the right number of strokes that could show a difference in MPs. Indeed, a statistically significant difference was detected between MPs at 20 and 30 cycles, making it clear that more masticatory cycles are not necessary. Furthermore, increasing the number of strokes after 50 caused a ceiling effect that did not allow for distinguishing between different MPs, and therefore, confirmed the previous theory.

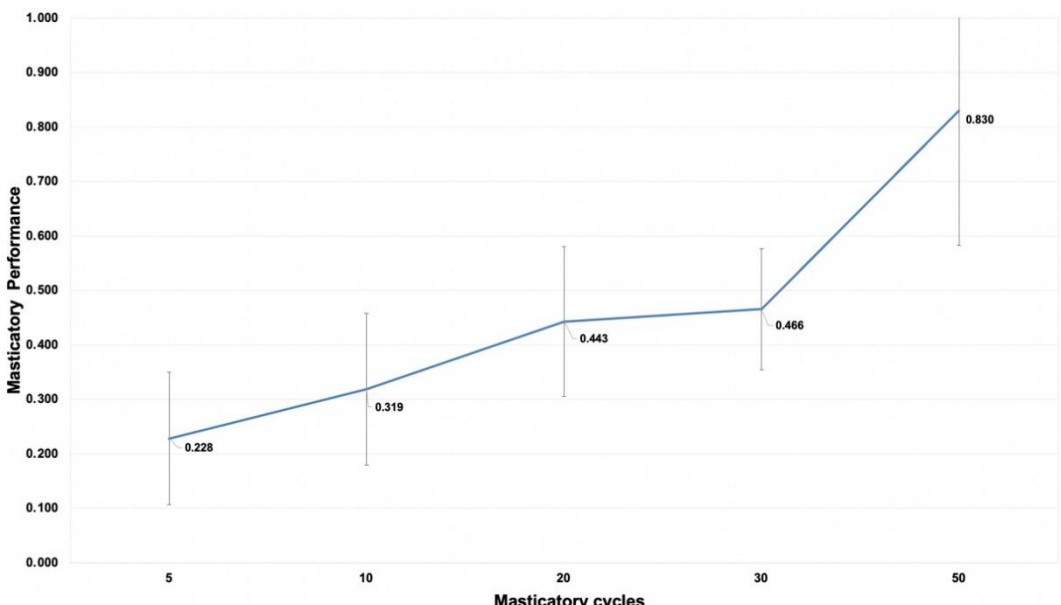

**Figure 4.** Means and standard deviations of MP in relation to the stepwise increase in the number of masticatory cycles.

According to Koo and Li, ICC scores showed a good correlation concerning both the intra- and inter-rater reliabilities and an excellent correlation between the results of the first test and those of

the second one [22]. In addition, a subjective evaluation of the specimens was performed in order to test the construct validity. Each specimen was classified into poorly, moderately, and highly mixed groups to verify whether the obtained MP values were consistent with the objective assessment. According to Prinz, the subjective assessment was coherent with the digital one, but errors can be introduced if a reference scale is not available and, of course, it cannot be as reliable and precise as the computer-assisted objective assessment of MP [10].

Overall, the results of the present study indicated that the proposed software was able to objectively and automatically assess MP. Nevertheless, this study has some limitations. The proposed method was compared neither to comminution tests nor to other chewing gum analyzer software. Moreover, the algorithm works in MATLAB. This could lead to complications if the user is not familiar with such a programming environment. Finally, low-quality images could have influenced the accuracy of the results of this study, thus not allowing for drawing conclusive statements regarding the reference MP values. However, the study aimed at describing and evaluating the validity of the novel proposed segmentation method as a proof of concept for the assessment of MP. Furthermore, the algorithm is freely available and, as the algorithm is pasted in MATLAB, the user only needs to upload bolus images. Finally, bolus images were captured using a prototypical tool aimed at standardizing the whole manipulation and acquisition procedure. Even if the used WiFi microcamera was not the most high-quality one, it was able to acquire images, from which, different MPs were assessed. Future research should pursue a double aim: (a) the standardization of the chewing gum manipulation procedure and the miniaturization of the equipment needed, and (b) the assessment of recognized MP values via the daily utilization of chewing mixing ability tests in clinical practice. Moreover, further research is needed to determine the validity and reliability of the proposed method in the assessment of MP in different dental statuses and in comparison to the gold standard tests.

## 5. Conclusions

Within the limitations of the present study, the proposed software allowed for analyzing the different MPs corresponding to different numbers of chewing cycles. This method provided an automatic identification of the colored areas, which were perfectly separated from the background of the images. It was demonstrated as being able to quantify the percentage of the mixed color area, providing quantitative data through the computerized analysis by using the best possible segmentation and minimizing human interaction.

**Author Contributions:** Conceptualization, G.R.; methodology, L.S.; software, R.N. and L.V.; formal analysis, L.A.; investigation, L.A.; data curation, M.M.; writing—original draft preparation, L.A. and M.M.; writing—review and editing, L.S., A.S. and G.R. All authors have read and agreed to the published version of the manuscript.

**Funding:** This research received no external funding.

**Acknowledgments:** The authors wish to thank Andrea Centoscudi, Luca Carloni, and Maria Chiara Cesaroni for their kind assistance and support, as well as all the students of the Polytechnic University of Marche Dental School for participating in this study voluntarily. The proposed algorithm may be requested by mailing the corresponding author.

**Conflicts of Interest:** The authors declare no conflict of interest.

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
