# Peer review of "A Novel Color-Based Segmentation Method for the Objective Measurement of Human Masticatory Performance"

_applsci, doi:10.3390/app10238626_

Round 1
Reviewer 1 Report
The aim of this study is to provide a novel method for the colour-mixing test as an objective measurement of human masticatory performance. I have several concerns about this study.
- Many novel and digital methods have been developed already to measure colour-mixing test. The authors need to make a solid justification why we need another novel method.
- The authors used subjective assessment as one of the validity tests. Since there are several gold-standard analyses (mentioned in the introduction), the validity test should be performed between the proposed novel test and gold-standard test.
- Sample size calculation is required for the validity and reliability tests.
- It is not clear that what are the results generated by the software. Please clarify what the numbers in the results stand for.
Author Response
Reviewer 1
We thank the reviewer for the constructive comments and suggestions. The manuscript has been revised according to the suggested modifications (highlighted in light blue).
- We thank the reviewer for this comment. Software may introduce errors due to a manual resize and segmentation of the boluses’ images and to a difficulty in distinguishing hue variation between the gum and the background, especially in well-mixed chewing-gums. Extracted data highly depend on the accuracy of the segmentation process, which is an essential step in computer vision and in automatic pattern recognition based on image analysis. Clustering systems could provide a high discriminative power of different color regions in each image using color information. This could allow to quantify the degree of color mixing depending on the number of chewing cycles or different dental patterns (lines 77-84, p.2).
- The Authors have tested construct validity using a visual inspection as suggested by Montero et al. 2020 (line 162, p. 5).
The aims of the study were to propose and describe a novel algorithm and to verify if the proposed method was able to distinguish the different masticatory performances at a different number of masticatory cycles. We agree with the reviewer that there are several gold-standard analyses: in a further phase of our project, we will use such tests in order to assess the validity of the method (lines 267-269, p. 8).
- No sample size calculation was made, but a sample size 2.5-fold greater than the one of the studies of Schimmel et al. (2007) and Yousof et al. (2019) was enrolled (lines 96-97, p.3).
- The algorithm automatically analyzes the images and gives as output the percentage of the mixed portion of the sample (lines 121-123, pp. 3-4).
Reviewer 2 Report
Dear authors congratulations form your manuscript. In my opinion it is a well-designed study and the materials and methods and results are well described, however som minor corrections are needed before publication.
English grammar and syntaxes should be checked. English impersonal form in passed tense should be used along the manuscript.
Discussion should be deeply redacted and other studies results should be compared with you results discussion the differences or similarities found.
In my opinion conclusions should be shortened to make a strong conclusions,
Agins, congratulations for the work.
Regards,
Author Response
Reviewer 2
We thank the reviewer for the constructive and kind comments and suggestions. The manuscript has been revised according to the suggested modifications (highlighted in yellow).
- We thank the reviewer for this suggestion. The manuscript was read and corrected by a native English-speaking colleague before our re-submission.
- We have extended the Discussion of the present study. Moreover, limitations and future prospects were reported (lines 212-269, pp. 7-8).
- We agree with the reviewer: we have shortened the conclusion of the present study (lines 271-276, p.8).
Reviewer 3 Report
" A novel color-based segmentation method for the objective measurement of human masticatory performance”
This study is a useful study focusing on evaluating an automatic color-based segmentation method to separate mixed and unmixed colors of images derived from the application of the two-color chewing-gum mixing test. This manuscript is well written. However, there are corrections that are essential to meet the standard for publication. Please refer to the following comments.
1) Please change the diagram to convey this study to the reader in an easy-to-understand manner. This study is to classify gum colors using k-means clustering. Please add a diagram showing the flow of research below
- Please add a photo of two-color gum.
- Please add a photo of the gum after chewing
Please explain how to handle this gum. What happened to the attached saliva? Did you unify the shooting directions to capture the gum photos?
2) The authors explained the PC analyzed as follows.
"Processing phase, necessary to classify an image into different color groups. A PC (Intel ® Core TM i7-6700HQ, 2.6 GHz, 16.0 GB) with Windows 10 Home was used as electronic equipment in order to process the acquired data and MATLAB ® R2019b (MATLAB, MathWorks) software to enable color-based segmentation. "
However, Figure 2 shows a diagram of a Macintosh PC.
Please explain.
3) It is very useful to be able to objectively recognize masticatory efficiency using k-means clustering. But are there any ideas to improve the accuracy? I want to hear your opinion.
4) Please show the limitations and prospects of this study in the discussion section.
Author Response
reviewer 3
We thank the reviewer for the constructive and kind comments and suggestions. The manuscript has been revised according to the suggested modifications (highlighted in green).
- We have added a new Figure (Figure 1) in which the two-color gum and the specimens after chewing and the flattery are depicted (lines 114-118, p.3). Moreover, we have better and further described the whole process of manipulation of gums specimens (lines 106-123, p. 3).
- We thank the reviewer for this comment. We apologize for the inconvenient. We have changed Figure 2 (now Figure 3) according to what was stated in the text. MatLab is able to run both in Macintosh PCs and in Windows ones. The Authors used a Macintosh PC to write the manuscript and screenshotted the screen in order to graphically describe how the program works. Actually, all the measurements were performed used a Windows 10 Home PC (lines 152-155, p.5).
- We thank the reviewer for this comment. Actually, we used a novel hardware developed by our research group. It is a prototype. The goal of this novel tool is to standardize the manipulation and the image acquisition of chewing-gum boluses. We have decided to install on the top of this tool a wifi camera that is able to communicate directly with the PC or mobile phone of the user. As the tool is still a prototype, the used wifi micro-camera is not the most high-qualitative one. This could have led to deficiencies in the accuracy of the results. The improvement in the quality of the acquisition camera, will probably improve the accuracy of the outcomes too (lines 253-269, p.8).
- We have added the limitations of the present study and its future prospects (lines 264-269, p.8).
Round 2
Reviewer 1 Report
My comments have been well addressed by the authors. I don't have further comments.